# SPTBN1 Mediates the Cytoplasmic Constraint of PTTG1, Impairing Its Oncogenic Activity in Human Seminoma

**DOI:** 10.3390/ijms242316891

**Published:** 2023-11-29

**Authors:** Emanuela Teveroni, Fiorella Di Nicuolo, Edoardo Vergani, Alessandro Oliva, Emanuele Pierpaolo Vodola, Giada Bianchetti, Giuseppe Maulucci, Marco De Spirito, Tonia Cenci, Francesco Pierconti, Gaetano Gulino, Federica Iavarone, Andrea Urbani, Domenico Milardi, Alfredo Pontecorvi, Francesca Mancini

**Affiliations:** 1International Scientific Institute Paul VI, Fondazione Policlinico Universitario A. Gemelli IRCCS, 00168 Rome, Italy; ema.teveroni@gmail.com (E.T.); fiorella.dinicuolo@gmail.com (F.D.N.); alfredo.pontecorvi@policlinicogemelli.it (A.P.); mancini.chicca@gmail.com (F.M.); 2Division of Endocrinology, Fondazione Policlinico Universitario A. Gemelli IRCCS, 00168 Rome, Italy; edoardo.vergani@outlook.it (E.V.); alessandrooliva996@gmail.com (A.O.); emanuel.vi93@gmail.com (E.P.V.); 3Department of Neuroscience, Section of Biophysics, Università Cattolica del Sacro Cuore, 00168 Rome, Italy; giada.bianchetti@unicatt.it (G.B.); giuseppe.maulucci@unicatt.it (G.M.); marco.despirito@unicatt.it (M.D.S.); 4Fondazione Policlinico Universitario A. Gemelli IRCCS, 00168 Rome, Italy; 5Division of Anatomic Pathology and Histology, School of Medicine, Università Cattolica del Sacro Cuore, 00168 Rome, Italy; tonia.cenci@gmail.com (T.C.); francesco.pierconti@policlinicogemelli.it (F.P.); 6Department of Urology, Fondazione Policlinico Universitario A. Gemelli IRCCS, 00168 Rome, Italy; gaetano.gulino@policlinicogemelli.it; 7Department of Basic Biotechnological Sciences, Intensivological and Perioperative Clinics, Catholic University of Sacred Heart, Largo Vito, 00168 Rome, Italy; federica.iavarone@policlinicogemelli.it (F.I.); andrea.urbani@policlinicogemelli.it (A.U.); 8Clinical Chemistry, Biochemistry and Molecular Biology Operations (UOC), Agostino Gemelli Foundation University Hospital IRCCS, Largo Agostino Gemelli 8, 00168 Rome, Italy

**Keywords:** TGCTs, seminoma, PTTG1, SPTBN1, interactome, invasiveness

## Abstract

Seminoma is the most common testicular cancer. Pituitary tumor-transforming gene 1 (PTTG1) is a securin showing oncogenic activity in several tumors. We previously demonstrated that nuclear PTTG1 promotes seminoma tumor invasion through its transcriptional activity on matrix metalloproteinase 2 (*MMP-2*) and E-cadherin (*CDH1*). We wondered if specific interactors could affect its subcellular distribution. To this aim, we investigated the PTTG1 interactome in seminoma cell lines showing different PTTG1 nuclear levels correlated with invasive properties. A proteomic approach upon PTTG1 immunoprecipitation uncovered new specific securin interactors. Western blot, confocal microscopy, cytoplasmic/nuclear fractionation, sphere-forming assay, and Atlas database interrogation were performed to validate the proteomic results and to investigate the interplay between PTTG1 and newly uncovered partners. We observed that spectrin beta-chain (SPTBN1) and PTTG1 were cofactors, with SPTBN1 anchoring the securin in the cytoplasm. SPTBN1 downregulation determined PTTG1 nuclear translocation, promoting its invasive capability. Moreover, a PTTG1 deletion mutant lacking SPTBN1 binding was strongly localized in the nucleus. The Atlas database revealed that seminomas that contained higher nuclear PTTG1 levels showed significantly lower SPTBN1 levels in comparison to non-seminomas. In human seminoma specimens, we found a strong PTTG1/SPTBN1 colocalization that decreases in areas with nuclear PTTG1 distribution. Overall, these results suggest that SPTBN1, along with PTTG1, is a potential prognostic factor useful in the clinical management of seminoma.

## 1. Introduction

Testicular germ cell tumors (TGCTs) represent the most frequent cancer in men between 20 and 40 years old. Seminomas account for the most common subtype of TGCTs [1]. The majority of seminoma-affected patients display clinical stage I disease in which the prognosis is very good, with a global survival rate around 98%. Nevertheless, about 15–20% of patients show tumor relapse [2,3]. In stage I seminoma, tumor size is reported as the most reliable prognostic factor used to evaluate the risk of disease recurrence. However, to date, no established cut-off value for tumor size can be assumed to guide the decision on adjuvant treatment [4]. Radiotherapy and carboplatin have been the most common adjuvant treatment options for stage I testicular seminoma after radical orchiectomy. However, due to detrimental consequences of adjuvant therapies, many groups aim to investigate alternative strategies for a better tailoring of the treatment options [3]. Nowadays, active surveillance is a well-established option in the management of most patients, whereas additional treatments are reserved only for those subjects who develop relapse [4]. However, the strategy of surveillance needs a relatively long period of follow-up, without a reliable prognostic model to foresee the risk of relapse [4]. Rete testis invasion (RTI) seemed to play a role as a potential prognostic factor, but its role in seminoma is still debated [5]. The presence of neoplastic lymphadenopathies is crucial since it defines Stage II seminomas. Stage II seminomas require a far more complex therapeutic approach (radiotherapy versus lymph node dissection versus polychemotherapy) given the worse prognosis and the more frequent relapses compared to Stage I [6,7]. To date, very few groups have focused on evaluating the differences between the molecular hallmarks of Stage I and II seminomas [8,9]. Thus, there is a lack of reliable prognostic factors that would be helpful in tailoring a more precise clinical decision. We previously highlighted the eligibility of the pituitary tumor-transforming gene 1 (PTTG1) as a promising new prognostic factor for seminoma cancer progression. PTTG1 is a securin with well-known oncogenic activity in several tumor histotypes [10,11,12,13,14]. PTTG1’s contribution to tumor progression is attributed to both its promotion of genetic instability, due to altered sister chromatid segregation during mitosis, and its transcriptional activity on different targets, such as matrix metalloproteinase 2 (*MMP-2*) [15,16,17]. We previously reported that PTTG1’s oncogenic properties, specifically in seminoma, were strictly associated with its nuclear distribution [18,19]. Indeed, we demonstrated that PTTG1’s nuclear localization was more pronounced in the seminoma periphery, the area that is more inclined to be highly invasive [18]. Moreover, we further correlated the nuclear localization of PTTG1 with the aggressive phenotype in three different seminoma cell lines (JKT-1, SEM-1, and TCAM2) [19]. In particular, we found that this phenomenon was associated with the PTTG1-dependent transcriptional activation of the *MMP-2* gene as well as its transcriptional repression of E-cadherin (*CDH1*) through Zinc finger E-box binding homeobox 1 (ZEB1) [19,20], one of the master players in epithelial-to-mesenchymal transition (EMT) [21]. Since the three human seminoma cell lines showed different aggressive behaviours that correlated with PTTG1 nuclear levels, we speculated that this specific subcellular localization could mark distinct seminoma stages. In particular, TCAM2, bearing the lowest PTTG1 nuclear levels, showed a less invasive phenotype [19]. Intriguingly, we found that in TCAM2, PTTG1’s nuclear translocation was impaired even if forced by the overexpression of PTTG-binding factor (PBF), a well-known inducer of PTTG1’s shift to the nucleus [22]. This prompted us to identify the factors responsible for the progressive nuclear translocation of PTTG1 in order to uncover new players in seminoma cancer progression that may be useful for prognosis. To this aim, we made use of a proteomic approach using PTTG1 immunoprecipitation in seminoma cell lines. The “interactome” analysis revealed novel PTTG1 partners that were differentially bound to the securin. Of interest, we found that spectrin alfa/beta-chain (SPTAN1 and SPTBN1) interacted with PTTG1, specifically in TCAM2. Spectrins belong to a cytoskeletal protein family and are assembled to build an extensive heterodimeric filamentous network that functions as cytoplasmic scaffold for several proteins [23]. In mammals, spectrin is composed of modular structures of α and β subunits. These proteins are encoded by seven genes: *αI, αII, βI, βII, βIII, βIV,* and *βV*. The spectrin genes *SPTA1* and *SPTAN1* encode the αI and αII isoforms, respectively, while four different genes (*SPTB*, *SPTBN1*, *SPTBN2*, and *SPTBN4*) encode the different β spectrin isoforms (βI-IV). The α and β subunits of spectrins are assembled alongside each other in an antiparallel manner, forming bar-shaped αβ dimers [23]. Spectrins are highly modular proteins that contain many spectrin repeats (SRs) [24], actin-binding and Src homology 3 (SH3) domains, and many other functional motifs that contribute to the multiple functions of spectrin [25,26]. Particularly, βII spectrin (SPTBN1) has been implicated in diseases such as cancer, neurological disorders, and cardiovascular diseases [27,28,29]. Of note, it has been reported that βII spectrin expression levels and functions are associated with different tumor stages or types [23,28]. Interestingly, a recent report by Wu and colleagues demonstrated that lower levels of SPTBN1 were correlated with shorter progression-free survival in breast cancer [30]. Furthermore, the authors showed that SPTBN1 is able to retain NF-κB p65 in the cytoplasm and that SPTBN1 loss promotes p65 nuclear translocation, which mediates the EMT process. Accordingly, in hepatocellular carcinoma, it was reported that the inhibition of SPTBN1 mediates β-catenin nuclear translocation, which was correlated with a less differentiated state of the tumor cells [31]. Therefore, we formulated the hypothesis that SPTBN1 could function as a cytoplasmic anchor for PTTG1, impairing its nuclear-associated oncogenic activity. This prompted us to investigate the functional interplay between SPTBN1 and PTTG1 in seminoma tumors.

## 2. Results

### 2.1. Interactome Analysis of PTTG1 Protein in Seminoma Cell Lines

To find out the differential partners involved in the subcellular localization of PTTG1, we performed immunoprecipitation of the securin from the lysates of multiple seminoma cell lines, including JKT-1, SEM-1, and TCAM2. The proteins that were co-immunoprecipitated with PTTG1 were separated by SDS–PAGE, and the bands that were obtained underwent MS proteomic analysis. Figure 1a shows a specific band at 250 kDa exclusively in the TCAM2 cell line on a coomassie stained gel. Among the proteins identified, we found, with high confidence, Spectrin-αII-chain (SPTAN1, Uniprot code Q13813) and Spectrin-βII-chain (SPTBN1, Uniprot code D6W5C0). Appendix A list the tryptic unique peptides related to the proteins SPTAN1 and SPTBN1, respectively, in the TCAM2 co-immunoprecipitated lysate. Figure 1 reports the protein coverage obtained by the combination of the tryptic peptides of SPTAN1 (Figure 1b) and SPTBN1 (Figure 1d) and the deconvoluted ESI spectra of representative tryptic fragment peptides related to these proteins, respectively (Figure 1c,e).

### 2.2. Differential SPTBN1/PTTG1 Binding in Seminoma Cell Lines

To validate the interaction between PTTG1 and spectrins and their differential reciprocal binding in JKT-1, SEM-1, and TCAM2 seminoma cells, we performed PTTG1 immunoprecipitation and immunofluorescence analysis. We selected SPTBN1 as a subunit of the spectrin complex because of its involvement in cancer [28]. A Western blot of total cell lysates revealed comparable SPTBN1 levels between the three seminoma cell lines (Figure 2a). As expected, co-IP analysis of PTTG1 uncovered differing SPTBN1 interactions, with TCAM2 bearing the highest binding capability (Figure 2a,b). This result validates the proteomic analysis, indicating that the PTTG1-SPTBN1 interaction is a prerogative of TCAM2 cells. Moreover, we used immunofluorescence analysis to further confirm the proteins’ colocalization. As reported in Figure 2c,d, the interaction of PTTG1 and SPTBN1 is different between the cell lines, with TCAM2 showing the highest levels of binding. Intriguingly, the increased binding of PTTG1 and SPTBN1 in JKT-1, SEM-1, and TCAM2 negatively correlates with the previously reported nuclear localization of PTTG1 and the aggressive behavior of the three cell lines [19].

### 2.3. Downregulation of SPTBN1 Facilitates PTTG1 Nuclear Localization in TCAM2 Cells

Since we uncovered SPTBN1 as a TCAM2-specific PTTG1 partner, we wondered if SPTBN1 could play a role in the differential subcellular localization of this securin. In fact, we formerly reported that TCAM2 retains the lowest nuclear localization of PTTG1 when compared to the other two seminoma cellular models [19]. In order to address the question, we downregulated SPTBN1 levels by siRNA in TCAM2 cells (Figure 3a). The efficiency of SPTBN1 silencing was about 60% (Figure 3a, right panel). Figure 3b,c show the immunofluorescence analysis of the subcellular localization of PTTG1 upon SPTBN1 silencing. As expected, the downregulation of SPTBN1 in TCAM2 cells led to a significant increase in PTTG1 in the nuclear fraction (Figure 3b,c). Furthermore, we analysed by western-blot the subcellular localization of PTTG1 following the treatment of TCAM2 cells with siRNA targeting SPTBN1 in combination with the overexpression of PBF, a well-known mediator of PTTG1’s nuclear translocation [22]. We found that SPTBN1 downregulation mediated a decrease in cytoplasmic PTTG1 localization along with a significant increase in its nuclear fraction (Figure 3d,e). All together, these findings support our hypothesis about the role of SPTBN1 as a PTTG1 interactor that has the specific ability to mediate PTTG1’s cytoplasmic restraint in TCAM2 cells.

### 2.4. PTTG1/SPTBN1 Binding Affects PTTG1 Nuclear Localization

To understand the PTTG1/SPTBN1 binding in more depth, we made use of a PTTG1 deletion mutant (PTTG1Δ159) lacking the SH3 interacting domain [32]. As shown in Figure 4a,b, PTTG1Δ159 binds SPTBN1 to a lesser extent (about 30%) in comparison to wild-type PTTG1 in TCAM2 cells. The subcellular fractionation revealed that wt-PTTG1 is mainly localized in the cytoplasm, as expected, whereas the mutant form was strongly localized in the nucleus (Figure 4c). Importantly, the quantification analysis showed a highly significant increase in the nuclear/cytoplasmic ratio (N/C) of the PTTG1-deletion mutant in comparison to the wild-type form (Figure 4d). Overall, these results further suggest a specific role for SPTBN1 in the cytoplasmic constraint of PTTG1.

### 2.5. Downregulation of SPTBN1 Promotes Invasiviness of TCAM2 Cells

To find out whether the observed PTTG1/SPTBN1 interaction involved in the maintenance of PTTG1 outside of the nucleus has a functional impact on TCAM2 cell behaviour, we used a 3D sphere-forming assay. As reported in Figure 5a,b SPTBN1 downregulation caused a significant increase in TCAM2 sphere-forming ability both alone and with, albeit slight, additive effect of PBF overexpression . This result confirms our previous finding about the crucial role that the nuclear localization of PTTG1 plays in promoting its oncogenic activity, which is prevented, at least in part, by its SPTBN1-mediated cytoplasmic restraint.

### 2.6. PTTG1/SPTBN1 Interplay in Human Seminoma Specimens

To uncover the clinical relevance of the interplay between PTTG1 and SPTBN1, we took advantage of the Atlas database [33]. Our analysis of SPTBN1 mRNA levels in non-seminoma (N-S; N = 65) and seminoma (S; N = 68) TGCT specimens revealed significantly lower levels of SPTBN1 mRNA in seminoma samples compared to non-seminomas (Figure 6a). This result, in light of the SPTBN1-mediated cytoplasmic anchoring of PTTG1, fits with our hypothesis that the nuclear localization of PTTG1 has a key role in promoting invasiveness, specifically in seminomas. Indeed, we previously reported that nuclear PTTG1 was a specific feature of the seminoma histotype [19]. Furthermore, we analyzed the colocalization of PTTG1/SPTBN1 in human specimens from patients who underwent therapeutic orchiectomy for seminomas. As shown in Figure 6b–d, the immunofluorescence analysis of SPTBN1/PTTG1 confirmed the colocalization of these proteins ex vivo and highlighted the negative correlation between PTTG1’s nuclear localization and its binding to SPTBN1. In particular, we evaluated the colocalization of these proteins in certain tumor areas that contain different subcellular distributions of PTTG1. In the areas where PTTG1 was predominantly cytoplasmic (zone 1), its colocalization with SPTBN1 reached 65%, whereas in the areas with a more pronounced nuclear localization of PTTG1 (zone 2), this percentage decreased significantly to 50% (Figure 6b–d).

## 3. Discussion

The findings of the present research shed light on the role of SPTBN1 as a PTTG1 interactor with the specific ability to mediate its cytoplasmic restraint. PTTG1 is a well-known oncogene involved in the progression of several cancer histotypes [10,11,12,13,14]. We previously demonstrated its contribution in seminoma tumors, showing that PTTG1’s oncogenic activity was associated with its nuclear localization [18,19]. In particular, we firstly reported that PTTG1’s nuclear localization was more pronounced in the seminoma periphery, which is a more invasion-prone area [18]. Thereafter, we made use of *in vitro* cellular model of seminoma to demonstrate the correlation between nuclear PTTG1 and a more aggressive phenotype in the JKT-1, SEM-1, and TCAM2 cell lines. This phenomenon was associated with two different molecular mechanisms. First, we showed that the invasive behavior of seminoma cells depends on the PTTG1-mediated transcriptional activation of the *MMP-2* gene [19]. Moreover, we demonstrated PTTG1’s oncogenic activity resides also in its capability to act as a transcriptional repressor. Indeed, we evidenced the interplay between PTTG1 and Zinc finger E-box binding homeobox 1 (ZEB1) [20] in the transcriptional repression of E-cadherin (*CDH1*), a well-known *primum movens* of epithelial-to-mesenchymal transition (EMT) [21]. The in vitro human seminoma cellular model represented by the three cell lines (JKT-1, SEM-1, and TCAM2) showed a decreasing gradient of nuclear PTTG1, and this feature correlates with the aggressive behavior of the cell lines [19]. In particular, TCAM2 showed the least nuclear localization of the securin and exhibited the least aggressive phenotype among the cell lines [19]. Moreover, in TCAM2 cells, PTTG1’s nuclear translocation was hindered even in the presence of PBF overexpression, which is usually able to mediate its nuclear localization [22]. This led us to hypothesize the existence of specific factors responsible for PTTG1’s subcellular localization. Here, we show for the first time through “Interactome” analysis that the spectrins SPTBN1 and SPTAN1 act as novel PTTG1 interactors. These proteins belong to a cytoskeletal family organized in heterodimeric filaments composed of αβ dimers [23]. Spectrins are involved in multiple cellular functions and act as molecular scaffolds for several protein complexes [23]. SPTBN1, through associated binding partners, stabilizes target proteins linked to multiple molecular signatures, including cell cycle, DNA repair, and TGF-β signaling [34,35,36,37]. In this regard, these cytoskeletal filaments are known to be multifunctional scaffold platforms located both in the nuclear and cytoplasmic membranes. Such roles led to the concept of the “Spectrinome”, which is useful in describing the plethora of spectrin interactors [38]. Moreover, the dysfunction of spectrins has been linked to several pathologies, including anemia, elliptocytosis, neurodegenerative diseases, and cancer [25,26,27,28,39]. The role of SPTBN1 has been reported in multiple tumors, such as ovarian carcinoma, hepatocellular carcinoma (HCC), and kidney tumors [31,40,41]. Of note, mounting evidence revealed differential SPTBN1 expression depending on tumor stage, with reduced SPTBN1 frequently associated with cancer progression in different tumor histotypes [31,40,41,42]. In particular, Tang and colleagues pointed out that SPTBN1/Smad4/TGF β signaling acts as a suppressor of colorectal cancer progression [43]. Furthermore, some studies have already evaluated the role of spectrins as possible markers of a response to antineoplastic chemotherapy. In this regard, in ovarian cancer, SPTAN1 expression increased after successful chemotherapy [44]. Moreover, the enhancement of SPTBN1 expression in chimeric antigen receptor T cells showed promising results in improving tumor-specific homing and therapeutic responses [45]. Intriguingly, one of the molecular mechanisms underlying the protective effect of SPTBN1 against cancer progression depends on its ability to restrain oncogenic factors and prevent their nuclear translocation. In particular, in breast cancer, SPTBN1 has been reported to prevent the nuclear translocation of NF-κB p65, which impairs EMT process [30]. As well, SPTBN1 hindered β-catenin’s nuclear translocation in hepatocellular carcinoma [31].

We therefore wondered if SPTBN1 could act as a cytoplasmic scaffold of PTTG1 in seminoma and be one the factors responsible for its subcellular localization. Of note, in our seminoma cellular models, we found that PTTG1 and SPTBN1 interact almost exclusively in TCAM2 cells in which PTTG1 is more cytoplasmic in comparison to the other cell lines. Moreover, the downregulation of SPTBN1 in TCAM2 led to an increase in the securin in the nucleus, supporting our hypothesis that SPTBN1 is a cytoplasmic anchor of PTTG1. To deepen the molecular interaction between PTTG1 and SPTBN1, we made use of a PTTG1 mutant with a C-terminal deletion that lacks the SH3-binding domain (PTTG1Δ159). The SH3 motif is a well-known protein–protein interaction domain that is able to mediate the assembly of multiprotein complexes [46]. This domain is present in spectrin α/β dimers, particularly the α subunit [23]. We found that the binding of the PTTG1-deletion mutant to SPTBN1 was significantly lower in comparison to the wild-type. Importantly, PTTG1Δ159 was strongly localized in the nucleus, supporting the role of SPTBN1 as a dampening platform for PTTG1’s nuclear translocation. Moreover, we demonstrated that this phenomenon has a functional impact on seminoma tumor progression; indeed, in the 3D sphere-forming assay, we found that the downregulation of SPTBN1 led to an increase in the diameter of TCAM2 spheres, which was induced by PTTG1 nuclear translocation. The Atlas database interrogation revealed that in human seminoma specimens, SPTBN1 levels were significantly lower in comparison to non-seminoma testicular tumors. This result fits with our previous findings showing a higher PTTG1 nuclear localization in seminoma compared to other TGCTs. In order to translate these results to clinical practice, we verified the interplay between PTTG1 and SPTBN1 in human seminoma specimens from patients who underwent therapeutic orchiectomy. To this aim, we evaluated the subcellular distribution of these proteins through confocal microscopy, and *ex vivo*, we confirmed that the colocalization of PTTG1/SPTBN1 occurs to a significantly higher extent in the area of the seminoma showing greater cytoplasmic PTTG1 levels. Taken together, these findings led us to hypothesize a model in which spectrins retain PTTG1 outside of the nucleus, thereby impairing its oncogenic activity in seminoma. This PTTG1/SPTBN1 interaction could be negatively affected during seminoma tumor invasion, and we reported a progressive PTTG1 nuclear localization in this setting. Different molecular mechanisms may be responsible for the displacement of PTTG1 and SPTBN1. During seminoma progression, SPTBN1 downregulation could occur, as reported for different cancer histotypes [31,40,41,42]. Moreover, this phenomenon could also be mediated by post-translational modifications (PTMs). Indeed, it has been reported that the phosphorylation of PTTG1 affects its subcellular localization by inducing its nuclear translocation [47,48]. On the other hand, it has been extensively reported that SPTBN1 mutations affect its ability to bind with specific partners. In particular, Cousin and colleagues demonstrated that different pathogenic SPTBN1 variants led to autosomal dominant neurodevelopmental syndrome [49]. These SPTBN1 variants affect the protein’s stability, disrupt its interactions with key molecular partners, and disturb cytoskeleton organization [46]. Moreover, taking advantage of next generation sequencing and bioinformatics tools, many studies correlated spectrin mutations with different pathologies, such as nervous system diseases and cancer [50]. Thus, it is important to deepen our knowledge about the molecular mechanisms underlying the interaction between PTTG1 and SPTBN1 in order to develop therapeutic strategies to stabilize and maintain PTTG1/SPTBN1 binding.

Regardless, the significance of our research lies in the identification of SPTBN1 as a factor that is able to impair the nuclear oncogenic activity of PTTG1 in seminoma. The main limitation of our study is that we made use of an *in vitro* model of seminoma tumors and confirmed the results in a few human seminoma specimens. Therefore, to highlight the potential clinical impact of our results, it is crucial to evaluate SPTBN1 expression levels and PTTG1 subcellular localization, along with their interaction, in different stages of seminoma to correlate these proteins’ crosstalk with tumor progression and the risk for disease relapse.

## 4. Materials and Methods

### 4.1. Proteomic Analysis

We performed immunoprecipitation of PTTG1 using the lysates from the seminoma cell lines JKT-1, SEM-1, and TCAM2 to identify new partners involved in the subcellular localization of the securin. The PTTG1 co-immunoprecipitated proteins were separated by SDS–PAGE, and the bands obtained underwent MS proteomic analysis. To perform the in-gel tryptic digestion, protein lines were excised from preparative gels after EZ Blue gel staining (Merck, Darmstadt, Germany). The in-gel digestion was carried out using “Trypsin Singles Proteomics Grade Kit” (Merck, Darmstadt, Germany). Tryptic digests were freeze-dried, solubilized in 0.1% formic acid (FA), and submitted to HPLC-ESI-high-resolution MS/MS experiments for protein identification.

For bottom-up RP-HPLC ESI-MS analyses, the chromatographic column used was EASY-Spray column 15 cm × 50 µm ID, PepMap C18 (2 µm particles, 100 Å pore size) coupled with Acclaim PepMap100 cartridge (C18, 5 μm, 100 Å, 300 μm i.d. × 5 mm) (Thermo Fisher Scientific, Waltham, MA, USA). The gradient elution was performed using eluent A (FA 0.1%, *v*/*v*) and solvent B (ACN:FA 99.9:0.1, *v*/*v*) and included the following steps: (i) 5% B (2 min), (ii) from 5% to 55% B (130 min), (iii) from 55% to 99% B (15 min), (iv) 99% B (10 min), (v) from 99% to 5% B (2 min), and (v) 5% B (13 min) at a flow rate of 0.3 μL/min. The injection volume was 5 μL, which corresponded with 0.25 μg of total protein concentration after opportune sample dilution with 0.1% (*v*/*v*) FA aqueous solution. Chromatographic separations were performed in triplicate at a thermostatic temperature of 40 °C. The Orbitrap Fusion Lumos instrument operated in positive ionization mode at a resolution of 1,200,000 in 350–1500 *m*/*z* scan filter range in Data-Dependent Scan (DDS) mode and performed MS/MS fragmentation using High Collision Dissociation (HCD) of the most intense signals of each full scan MS spectrum. The minimum signal was set to 500.0 and the isolation width to 2.00 *m*/*z*. Normalized collision energy was set at 35.0. Capillary temperature was 250 °C, and the source voltage was +1.5 kV. MS/MS spectra acquisition was performed in the linear ion trap at normal scan rate. Acquisition started at 4 min in order to avoid salt contamination from the source during the first minute of elution. All samples were analyzed in triplicate. MS data were analyzed using the Thermo Scientific Proteome Discoverer (PD) 2.4 software, with SEQUEST HT database search algorithms (University of Washington, Seattle, WA, USA, licensed by Thermo Electron Corp, Waltham, MA, USA).

### 4.2. Human Seminoma Specimens

This research was carried out in accordance with the guidelines of the Declaration of Helsinki. All patients signed an informed consent for research purposes. Ethics committee of the Fondazione Policlinico Universitario “A. Gemelli”, Rome, Italy approved the protocol (ID 4824). The inclusion criteria were age ranging from 20 to 50 years and diagnosis of pure seminoma with ≥5 mm diameter. The specimens were derived from testicular tissues of three patients who underwent an orchiectomy for seminoma at the Department of Surgical Pathology Fondazione Policlinico Universitario “A. Gemelli”. Confocal microscopy analysis was performed on formalin-fixed and paraffin-embedded tissues. After rehydration, the deparaffinized tissue slides were treated with 10 mM citrate buffer at pH 6.0 for 10 min to retrieve the antigen. After washing procedures, all primary antibodies were incubated for 1 h at RT. The primary antibodies used were rabbit monoclonal PTTG1 (Abcam, Cambridge, UK, 1:500) and mouse monoclonal SPTBN1 (Santa Cruz, Santa Cruz, CA, USA, 1:50). The PTTG1 was revealed using the Alexa Fluor 488-conjugated goat anti-Rabbit IgG secondary antibody (ThermoFisher Scientific, Waltham, MA, USA, 1:1000 for 1 h). Then, the slides were washed in PBS and then incubated with mouse monoclonal SPTBN1. This antibody was visualized using the Alexa Flour 594-conjugated goat anti-Mouse IgG secondary antibody (ThermoFisher Scientific, Waltham, MA, USA, 1:1000 for 1 h). After rinsing, the slides were finally mounted in Vectashield (H-1000, Vector Laboratories, Peterborough, UK), and double immunofluorescence slides were analyzed under confocal immunofluorescence microscopy.

### 4.3. Cell Culture and Transfections

JKT-1, SEM-1, and TCAM2 seminoma cell lines (kindly provided by Dr. A. L. Epstein) and 293T (kindly provided by Dr. F. Moretti) were cultured in RPMI containing 10% FBS (Millipore, Burlington, MA, USA) stable glutamine (glutamax, Thermofisher, Waltham, MA, USA), and penicillin/streptomycin mix solution (Thermofisher, Waltham, MA, USA). Transient transfections were carried out using Jet Prime Polyplus according to the manufacturer’s instructions (Polyplus, Illkirch-Graffenstaden, France). Plasmids used were pUHD-FLAG-PTTG1, pUHD-FLAG-Δ159-PTTG1 (kindly provided by Dr. Randy Y.C. Poon), pCIneo HA-PBF, SPTBN1 siRNA (siSPTBN1), and control siRNA (siCTL), which were purchased from Thermofisher (Stealth RNAi). Gene silencing was performed using RNAiMAX reagent according to the manufacturer’s instructions (Thermofisher, Waltham, MA, USA).

### 4.4. Immunofluorescence

For immunofluorescence analysis, we first fixed seminoma cells with 3.7% formaldehyde for 15 min at RT, then permeabilized them with 0.05% Triton X-100 in PBS, and blocked with 5% bovine serum albumin (BSA) for 1 h at RT. Cells were incubated with primary antibodies (rabbit αPTTG1, Abcam, Cambridge, UK; mouse αSPTBN1, Santa Cruz, Santa Cruz, CA, USA) and then incubated with goat Alexa Fluo-488 α-rabbit IgG and/or goat Alexa Fluo-594 α-mouse IgG (Molecular Probes, Eugene, OR, USA).

### 4.5. Confocal Microscopy and Colocalization Analysis

Cells were imaged with an inverted confocal microscope (Nikon A1-MP). Fluorescence images were collected on three separated channels (excitation: 402 nm, emission: 450/50 nm for the blue channel; excitation: 488 nm, emission: 525/50 nm for the green channel; excitation: 561 nm, emission: 595/50 nm for the red channel) using a 60X oil-immersion objective with 1.4 numerical aperture (NA). Internal photon multiplier tubes collected images with pixel resolutions of 1024 × 1024 in 16 bit at 0.063 ms dwell time. To evaluate colocalization, the Colocalization Threshold plugin available in ImageJ (NIH) was used. The Threshold Manders’ coefficients (tM1 and tM2) were calculated with automatic threshold settings defined by Costes regression approach [51]. Colocalization of PTTG1 and SPTBN1 in cell lines was evaluated through the quantification of tM1. For each cell line, three fields were considered (n = 90 for JKT-1, n = 70 for SEM-1, and n = 50 for TCAM2). The colocalization of PTTG1 and DAPI was quantified as previously reported [19]. Ten different fields were considered for each RNA interference treatment (n = 100 for iCTR, n = 150 for iSPTBN1). For evaluating the colocalization of PTTG1 and SPTBN1 in seminoma specimens, two different areas of the sample were considered, including one with predominantly cytoplasmic PTTG1 (Zone 1) and the other with more noticeable nuclear PTTG1 localization (Zone 2). For each zone, five fields were counted (n = 150 for zone 1, n = 200 for zone 2), and tM2 was quantified.

### 4.6. Immunoprecipitation and Western Blot Analysis

For immunoprecipitation (IP), 293T cells were lysed in IP lysis buffer (50 mM Tris–Cl, pH 7.5, 150 mM NaCl, 1% Nonidet P-40, 1 mM EDTA) supplemented with a mix of protease inhibitors (Boehringer, Ingelheim am Rhein, Germany). For IP α-FLAG, lysates were incubated overnight with the M2 αFLAG antibody at 4 °C. In order to isolate the immunocomplexes, we made use of Dynabeads according to the manufacturer’s instructions (ThermoFisher, Waltham, MA, USA). For interactome experiments in JKT1, SEM-1, and TCAM2, we made use of Tosyl-activated Dynabeads (Dynabeads M-280 Tosylactivated, Invitrogen, Waltham, MA, USA). Firstly, we performed the crosslink between the Dynabeads and PTTG1 antibody at 37 °C overnight. Then we performed the PTTG1 immunoprecipitation overnight at 4 °C using the three seminoma cell lysates (lysis buffer: Hepes 20 mM, NaCl 250 mM, EDTA 0.5 mM, Nonidet P-40 0.25%, plus a mix of protease inhibitors). For Western blot analysis, cells were lysed in RIPA buffer (50 mM Tris–Cl, pH 7.5, 150 mM NaCl, 1% Nonidet P-40, 0.5% Na deoxycholate, 0.1% SDS, 1 mM EDTA) or in the above-mentioned IP buffers containing a mix of protease inhibitors (Boehringer, Ingelheim am Rhein, Germany). Proteins were separated using SDS–PAGE and then blotted onto PVDF membranes (Millipore, Burlington, MA, USA), which were developed using enhanced chemiluminescence (ECL westar, Cynagen, Bologna, Italy). The band analysis was carried out using the Alliance 2.7 (UVITEC, Cambridge, UK) chemiluminescence imaging system, and the relative densitometry was quantified using the Q9 software Alliance V_1801. The following primary antibodies were used: rabbit αPTTG1 (Abcam, Cambridge, UK), mouse αHA (BioLegend, San Diego, CA, USA), mouse αSPTBN1 (Santa Cruz, Santa Cruz, CA, USA), mouse αSP1 (Santa Cruz, Santa Cruz, CA, USA), mouse αFLAG M2 (Merck, Darmstadt, Germany), mouse α-tubulin (Merck, Darmstadt, Germany). Tubulin was used as whole cell lysate and cytoplasmic loading control, whereas Sp1 was used as nuclear loading control.

### 4.7. Isolation of Nuclear/Cytoplasmic Fractions

In order to isolate the nuclear and cytoplasmic fractions, the cells were scraped off the plate with PBS and resuspended in hypotonic lysis buffer (10 mM HEPES pH 7.9, 10 mM KCl, 0.1 mM EDTA, 0.1 mM EGTA) supplemented with protease inhibitors (Boehringer) for 15 min at 4 °C. Then, 0.6% NP-40 was added, and the nuclei were isolated by centrifugation at 300× *g* for 5 min at 4 °C. After the centrifugation, we obtained the cytoplasmic supernatant and nuclei pellet. The nuclei were then resuspended in nuclear extract buffer (20 mM HEPES pH 7.9, 25% glycerol, 0.4 M NaCl, 0.1 mM EDTA, 0.1 mM EGTA), sonicated three times for 5 s at 20% amplitude, and finally centrifuged at 15,000× *g* for 5 min at 4 °C to recover the nucleoplasms.

### 4.8. Sphere-Formation Assay

TCAM2 cells were transfected with PBF or control vector in combination with the siRNA of SPTBN1 or control stealth RNA. After 24 h, for spheroid formation, 1000 TCAM2 cells/well were seeded in a 96-well, ultra-low-attachment (ULA) multiplate and diluted in 200 µL of medium in the presence of 20% methylcellulose (MC) to promote intercellular aggregation. Cells were imaged using phase-contrast microscopy after 4 days (Leica Microsystems, magnification 10×).

### 4.9. Statistical Analysis

Data are presented as values of mean ± standard deviation (SD). Paired two-tailed Student’s *t*-test was used as statistical test, and *p* < 0.05 was considered significant. The Atlas database was analyzed using unpaired *t*-test with Welch’s correction (that is, not assuming equal SDs) and GraphPad Prism 7.04 software (San Diego, CA, USA). In the Atlas database, we performed the analysis on mRNA levels of SPTBN1 in non-seminoma (N-S; N = 65) and seminoma (S; N = 68) specimens [29]. Data were reported as FPKM (fragments per kilobase of exon model per million reads mapped), a unit used to estimate gene expression based on RNA-seq data.

## 5. Conclusions

The new findings of our research suggest that in human seminoma, SPTBN1 plays a key role as a cytoplasmic constraint for PTTG1, impairing its nuclear oncogenic activity. According to our model, the interaction between PTTG1 and SPTBN1 could be impaired during seminoma development, leading to a progressive PTTG1 nuclear localization. In this scenario, the loss (or the decrease) of SPTBN1 could represent the first step toward the acquisition of a more aggressive and invasive phenotype. Further studies in human seminoma tissues are needed to correlate the interplay of PTTG1/SPTBN1 with tumor staging in order to provide a translational impact for our data. Therefore, this would emphasize the eligibility of SPTBN1 as a novel prognostic factor, along with PTTG1, for the clinical management of seminoma tumors.

## Figures and Tables

**Figure 1 ijms-24-16891-f001:**
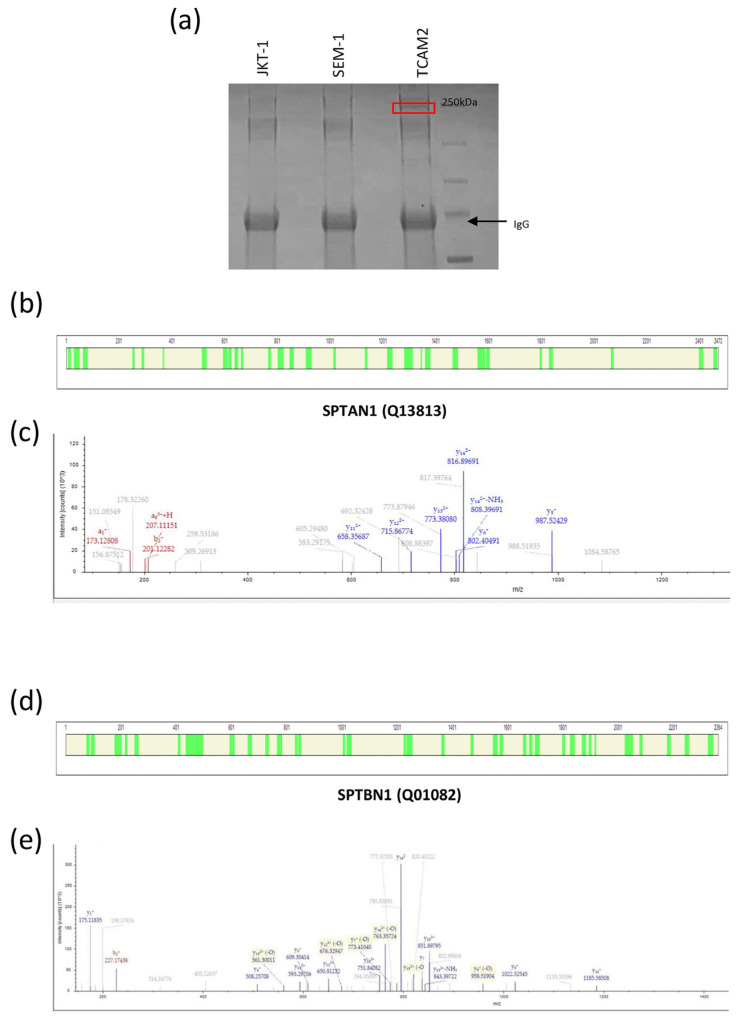
PTTG1 protein interactome analysis in JKT-1, SEM-1, and TCAM2 cell lines. (**a**) Coomassie gel staining of PTTG1 after immunoprecipitation in the JKT-1, SEM-1, and TCAM2 seminoma cell lines. The frame indicates the protein band selected for the subsequent proteomic analysis. The arrow indicates the IgG heavy chain. (**b**) Protein coverage obtained using the sum of the tryptic peptides for SPTAN1 protein, here represented in green. Protein and peptide spectra matches were validated by the calculation of false discovery rate (FDR) using the Percolator node. The strict target FDR value was set at 0.01. Protein identification results were filtered according to the Human Proteome Project Mass Spectrometry Data Interpretation Guidelines for identification of high-confidence peptides: peptide confidence was high; minimum peptide length was ≥9 amino acid residues; peptide rank was 1; and minimum was 2 peptides per protein. (**c**) Deconvoluted ESI spectrum of the tryptic fragment peptide with [M + H]1+ = 1745.8653 Da (z = +3, mono = 582.6266 Da RT = 15.47 min sequence: LSSDDNTIGKEEIQQR), which was related to the protein Spectrin alpha chain (Uniprot code Q13813) obtained with HCD fragmentation. LC-MS and MS/MS data were elaborated on using Proteome Discoverer 2.4 software (Thermo Fisher Scientific, Waltham, MA, USA) based on SEQUEST HT cluster as search engine against the Swiss-Prot Homo Sapiens proteome (UniProtKb, Swissprot, homo + sapiens released in September 2023). (**d**) Protein coverage obtained by the sum of the tryptic peptides for SPTBN1 protein, here represented in green. Protein and peptide spectra matches were validated by the calculation of false discovery rate (FDR) using the Percolator node. The strict target FDR value was set at 0.01. Protein identification results were filtered according to the Human Proteome Project Mass Spectrometry Data Interpretation Guidelines for identification of high-confidence peptides: peptide confidence was high; minimum peptide length was ≥9 amino acid residues; peptide rank was 1; minimum was 2 peptides per protein. (**e**) Deconvoluted ESI spectrum of the tryptic fragment peptide with [M + H]1+ = 1815.87503 Da (z = +3, mono = 605.96320 Da RT = 17.93 min sequence: ILSSDDYGKDLTSVMR M-15 Oxidation), which was related to the protein Spectrin beta chain (Uniprot code Q01082) obtained with HCD fragmentation. LC-MS and MS/MS data were elaborated on using Proteome Discoverer 2.4 software (Thermo Fisher Scientific) based on SEQUEST HT cluster as search engine against the Swiss-Prot Homo Sapiens proteome (UniProtKb, Swissprot, homo + sapiens released in September 2023).

**Figure 2 ijms-24-16891-f002:**
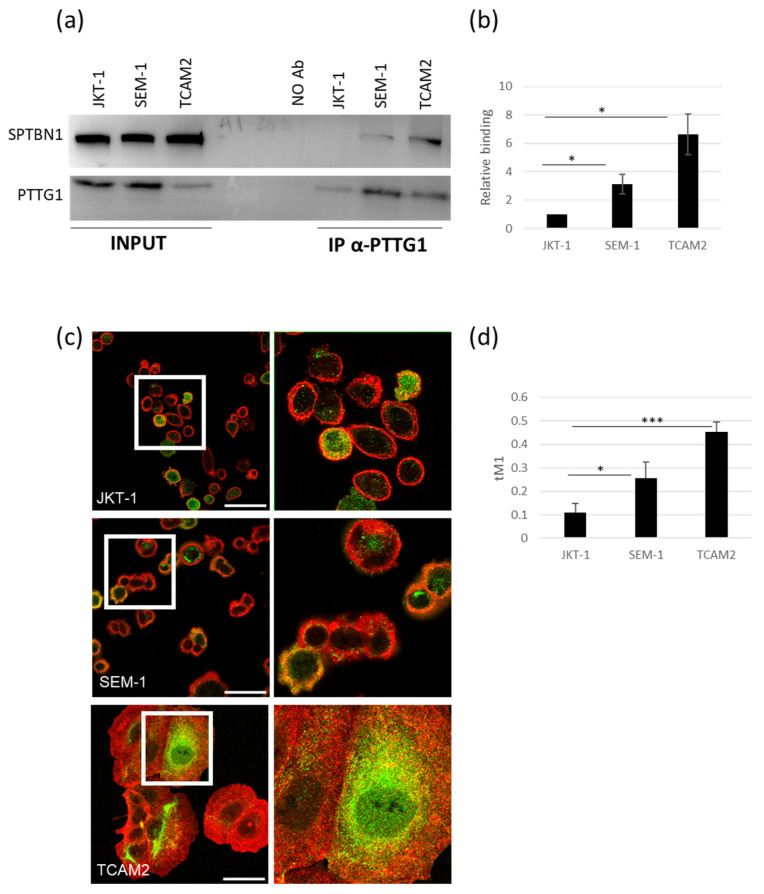
Analysis of PTTG1/SPTBN1 binding in seminoma cell lines. (**a**) Representative Western blot analysis of the indicated proteins in whole cell extracts (INPUT) of the three seminoma cell lines and the relative PTTG1 immunoprecipitation (IP α-PTTG1). (**b**) Densitometric analysis of the PTTG1/SPTBN1 relative binding. The interaction level between the two proteins in JKT-1 cell extracts was arbitrarily set to 1 (N = 3, * = *p* < 0.05, two-tailed unpaired *t*-test). (**c**) Representative pictures of merged confocal immunofluorescence analyses of PTTG1 (green) and SPTBN1 (red) in JKT-1, SEM-1, and TCAM2 cells. The PTTG1-SPTBN1 colocalized pixels are shown in yellow. Scale bar: 50 µm. White boxes represent the selected area reported with higher magnification on the right panels (**d**) Histogram reports the value of tM1 colocalization coefficient of PTTG1-SPTBN1, expressed as a percentage. For each cell line, three fields were counted (n = 90 for JKT-1, n = 70 for SEM-1, and n = 50 for TCAM2; * = *p* < 0.05, *** = *p* < 0.001, two-tailed unpaired *t*-test).

**Figure 3 ijms-24-16891-f003:**
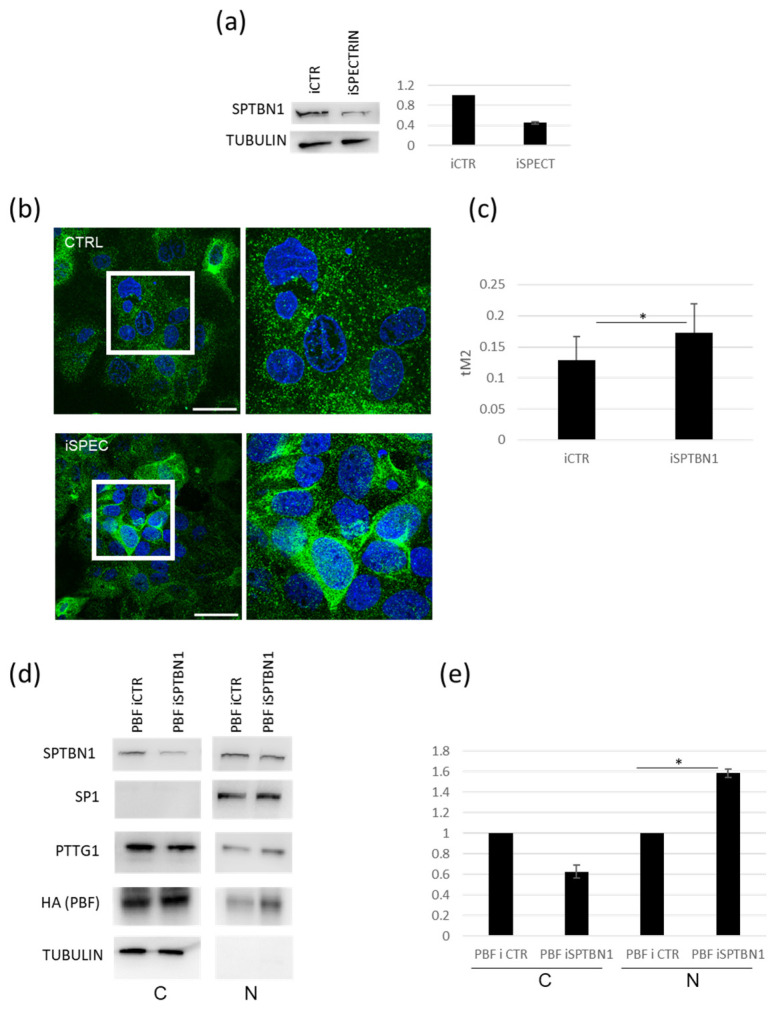
Modulation of the influence of SPTBN1 on PTTG1’s nuclear localization in TCAM2 cells. (**a**) Representative Western blot analysis of the indicated proteins in TCAM2 cells with control (iCTR) or SPTBN1 (iSPTBN1) RNA interference (left panel) and relative densitometric analysis (right panel) of two independent experiments in which iCTR was arbitrarily set to 1. (**b**) Representative pictures of merged confocal immunofluorescence analyses of PTTG1 (green) and nuclei (counterstained with DAPI, here shown in blue) in TCAM2 cells with iCTR or iSPTBN1. White boxes represent the selected area reported with higher magnification on the right panels. Scale bar: 50 µm. (**c**) Histogram reports the value of tM2 colocalization coefficient of PTTG1-DAPI, expressed as a percentage. For each RNA interference treatment, 10 fields were counted (n = 100 for iCTR, n = 150 for iSPTBN1; * = *p* < 0.05, two-tailed unpaired *t*-test). (**d**) Representative Western blot analysis of the indicated proteins in TCAM2 cells overexpressing PBF with iCTR or iSPTBN1 RNA interference. Cell lysates were fractionated in cytoplasmic (C) and nuclear (N) compartments. (**e**) Relative densitometric analysis of three independent experiments; iCTRs in both compartments were arbitrarily set to 1 (* = *p* < 0.05, two-tailed unpaired *t*-test).

**Figure 4 ijms-24-16891-f004:**
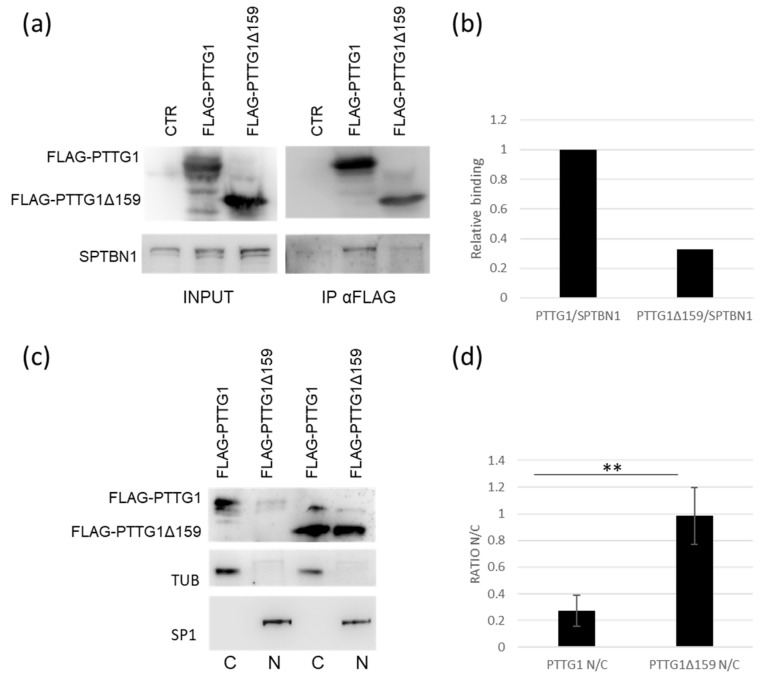
PTTG1/SPTBN1 binding affects PTTG1 nuclear localization. (**a**) Representative Western blot analysis of the indicated proteins in TCAM2 cells with overexpression of FLAG-PTTG1 or PTTG1-deletion mutant FLAG-PTTG1Δ159 in whole cell extracts (INPUT) and the relative PTTG1 immunoprecipitation (IP α-FLAG). (**b**) Densitometric analysis of SPTBN1 binding to FLAG-PTTG1 or PTTG1-deletion mutant FLAG-PTTG1Δ159. SPTBN1/PTTG1 interaction was arbitrarily set to 1. (**c**) Representative Western blot analysis of the indicated proteins in TCAM2 cells overexpressing FLAG-PTTG1 or PTTG1-deletion mutant. Cell lysates were fractionated in cytoplasmic (C) and nuclear (N) compartments. (**d**) N/C ratio of TCAM2 overexpressing FLAG-PTTG1 was arbitrarily set to 1. Relative densitometric analysis of three independent experiments (** = *p* < 0.01, two-tailed unpaired *t*-test).

**Figure 5 ijms-24-16891-f005:**
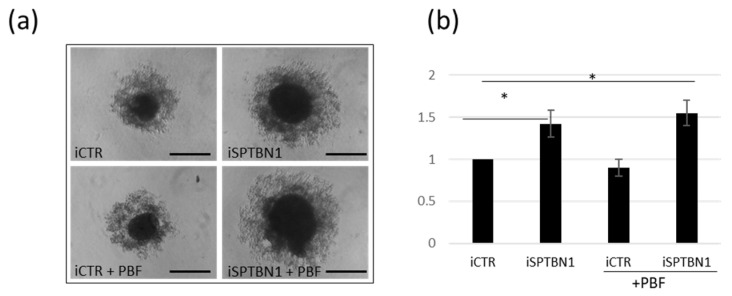
Downregulation of SPTBN1 promotes invasiveness of TCAM2 cells. (**a**) Representative pictures of spheroid cell formation assay of TCAM2 cells with control (iCTR) or SPTBN1 (iSPTBN1) RNA interference with or without PBF overexpression. Scale bar: 100 µm (**b**) Histogram shows changes in diameters of spheroids of SEM-1 cells with the indicated transfections 1 (N = 3, * = *p* < 0.05, two-tailed unpaired *t*-test).

**Figure 6 ijms-24-16891-f006:**
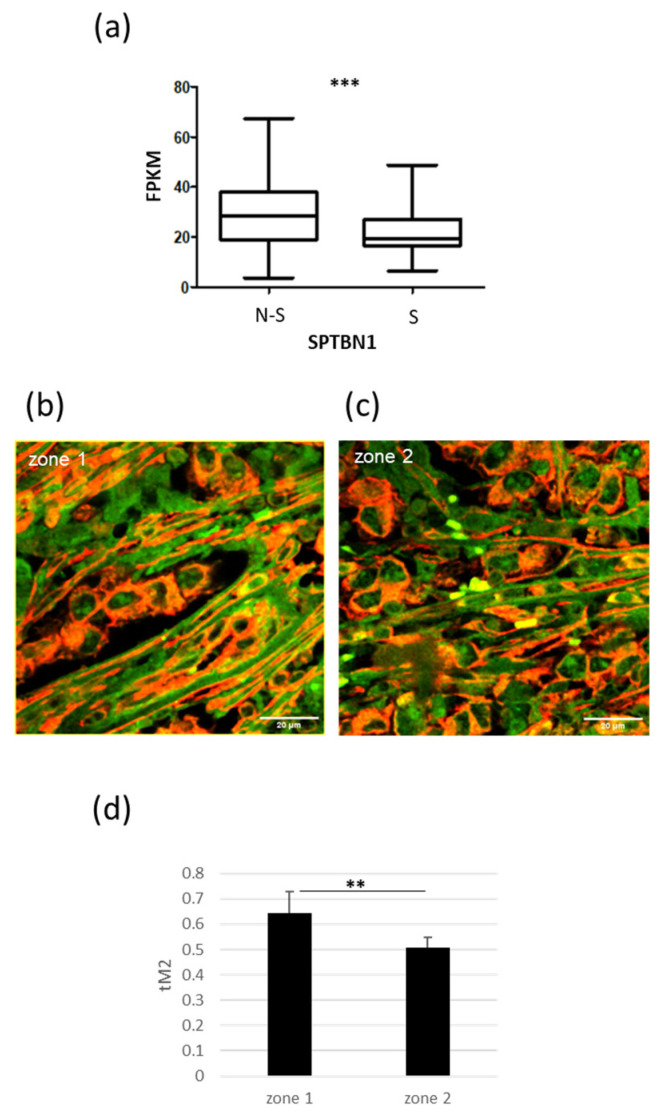
Interplay of PTTG1/SPTBN1 in human seminoma specimens. (**a**) Box plot of mRNA levels of SPTBN1 in non-seminoma (N-S; N = 65) and seminoma (S; N = 68) specimens in Atlas database (https://www.proteinatlas.org/ENSG00000115306-SPTBN1/pathology/testis+cancer (accessed on 8 September 2023) (*** = *p* < 0.001). (**b**,**c**) Representative pictures of merged confocal immunofluorescence analyses of PTTG1 (green) and SPTBN1 (red) in a human seminoma specimen. The PTTG1-SPTBN1 colocalized pixels are shown in yellow. Zone 1 indicates areas with predominantly cytoplasmic PTTG1. Zone 2 indicates areas with more noticeable nuclear PTTG1 localization. Scale bar: 20 µm. (**d**) Histogram reports the value of tM2 colocalization coefficient of PTTG1-SPTBN1, expressed as a percentage. For each zone five fields were counted (n = 150 for zone 1, n = 200 for zone 2; ** = *p* < 0.01, two-tailed unpaired *t*-test).

## Data Availability

The data presented in this study are available in this article (and Appendix A).

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
