# Peer review of "SPTBN1 Mediates the Cytoplasmic Constraint of PTTG1, Impairing Its Oncogenic Activity in Human Seminoma"

_ijms, 2023, doi:10.3390/ijms242316891_

Round 1
Reviewer 1 Report
Comments and Suggestions for Authors
The manuscript is good and discuss important issue but it needs some improvement:
1. In Methodology, I suggest to start with the section of human instead of the cell line.
2. In methodology: you mentioned the source of all used cell line except 293T, clarify?
3. What was the used reference in your western analysis?
4. What are your exclusion and inclusion criteria for the patients?
5. Clarify how many samples were collected from the patients.
6. Fig. 1, you need to enhance the resolution of all panels except a.
7. The discussion section is short and needs to give complete overview of the work.
Reviewer 2 Report
Comments and Suggestions for Authors
No comments. This is a well conducted investigation, with clearly presented and very interesting results. Nice piece of work.
Additional comments
Title: SPTBN1 mediates the cytoplasmic constraint of PTTG1 impairing its oncogenic activity in human seminoma. By Emanuela Teveroni et al.
The Article studied the relation between the spectrin SPTBN1 and the securin PTTG1 in seminoma tumours, with the hypothesis that SPTBN1 could function as cytoplasmic anchor for PTTG1, impairing its nuclear associated oncogenic activity. The results show a PTTG1/SPTBN1 co-localization in human seminoma specimen, that decreases in areas with nuclear PTTG1 distribution.
Abstract
1. Although the article proposes that SPTB1 might function as cytoplasmic anchor for PTTG1, impairing its nuclear associated oncogenic activity, a clear conclusion about the findings obtained is not given. So, it is not clear what it adds to the subject area.
Introduction
2. Despite all the information mentioned, it is not clear what is exactly the main question addressed by the research.
Material and methods
3. An experimental design making clear the sequence and support of experiments is missing.
4. A control using normal cells would be necessary to evaluate the interaction of SPTBN1 and PTTG1 in normal conditions.
5. Figure 2 captions should clearly describe what are we seeing in subsection c: Cells? Nuclei? It should be clearly indicated the distribution of SPTBN1 and PTTG1 with arrows.
6. Page 9 line 249. How were non-seminoma obtained? This is not clear and should be clearly indicated.
7. Figure 6a. It is not so clear the significant differences between N-S and S cells. This should be corroborated and explained in detail.
8. Figure 6b. It should be clearly indicated in the colocation of PTTG1 and SPTB1 in the cells. The pictures do not clearly show what is mentioned.
Discussion
9. A clear conclusion should be added at the end of the discussion section.
Reviewer 3 Report
Comments and Suggestions for Authors
Kindly refer to the attached report

Kindly refer to the attached report
Reviewer 4 Report
Comments and Suggestions for Authors
thank you for an interesting paper
great proposal for a research
introduction is well written
all data are clearly presented, nothing more to add
the final results has to be more highlighted
mayby a flow chart with inclusion/exclusion criteria for the study will be a valuable addition
thank you for an interesting paper
